# Gut Microbiome Alteration in HIV/AIDS and the Role of Antiretroviral Therapy—A Scoping Review

**DOI:** 10.3390/microorganisms12112221

**Published:** 2024-11-01

**Authors:** Zsófia Gáspár, Blin Nagavci, Bálint Gergely Szabó, Botond Lakatos

**Affiliations:** 1National Institute of Hematology and Infectious Diseases, Central Hospital of Southern Pest, H-1097 Budapest, Hungary; 2Doctoral School of Clinical Medicine, Semmelweis University, H-1097 Budapest, Hungary; 3Departmental Group of Infectious Diseases, Department of Internal Medicine and Hematology, Semmelweis University, H-1097 Budapest, Hungary

**Keywords:** AIDS, acquired immunodeficiency syndrome, HIV, human immunodeficiency virus, microbiome, dysbiosis, microbiota, metagenomics, non-nucleoside reverse transcriptase inhibitor, integrase strand transfer inhibitors

## Abstract

(1) Background: The gut microbiota plays a crucial role in chronic immune activation associated with human immunodeficiency virus (HIV) infection, acquired immune deficiency syndrome (AIDS) pathogenesis, non-AIDS-related comorbidities, and mortality among people living with HIV (PLWH). The effects of antiretroviral therapy on the microbiome remain underexplored. This study aims to map the evidence of the impact of integrase strand transfer inhibitors (INSTI) and non-nucleoside reverse transcriptase inhibitors (NNRTI) on the gut microbiota of PLWH. (2) Methods: A scoping review was conducted using PubMed, Web of Science, and Embase, with reports collected following PRISMA for Scoping Reviews (PRISMA-ScR). (3) Results: Evidence suggests that INSTI-based regimes generally promote the restoration of alpha diversity, bringing it closer to that of seronegative controls, while beta diversity remains largely unchanged. INSTI-based therapies are suggested to be associated with improvements in microbiota composition and a tendency toward reduced inflammatory markers. In contrast, NNRTI-based treatments demonstrate limited recovery of alpha diversity and are linked to an increase in proinflammatory bacteria. (4) Conclusions: Based on the review of the current literature, it is indicated that INSTI-based antiretroviral therapy (ART) therapy facilitates better recovery of the gut microbiome.

## 1. Introduction

### 1.1. Effects of HIV Infection on the Immune System and Intestinal Barrier

The gut microbiota is a complex system composed of 10 to 100 trillion bacterial cells, fungi, and viruses. It exists in a symbiotic relationship with the host, particularly the host’s immune system [1,2,3]. The functionality of the microbiota, and its communication with the host, is further supported by a network of metabolomic and proteomic interactions [4,5]. However, both host-related factors and environmental exposures can significantly influence its composition [6]. In the following paper, we will discuss the mechanisms by which HIV infection and ART treatment impact the gut microbiota.

HIV infection leads to a reduction of CD4 T cells in the gut-associated lymphoid tissue (GALT), a major site for HIV replication during the acute phase of infection, and is responsible for the majority of CD4 T cell depletion [7,8,9]. Subsets of CD4 T cells, such as T helper (Th) 17 and Th22 cells, are critical for maintaining the intestinal barrier and regulating antimicrobial defenses [10,11]. Disruption of immune function during HIV infection compromises the integrity of the intestinal barrier, leading to a breach between the intestinal lumen and systemic circulation. Increased intestinal barrier permeability permits the translocation of the microbiota and its components into the bloodstream, which triggers immune responses and initiates systemic inflammation. This disruption alters the composition of the microbiome, resulting in a state known as gut dysbiosis [9,12]. Microbial metabolites also enter the systemic circulation, further influencing immune function [13,14].

The elevated number and diversity of antigens in circulation activate the innate immune system through cytokine pathways, contributing to monocyte activation and improper functioning of gut-homing CD4+ T cells, which perpetuates immune activation [15]. This chronic immune activation has been linked to the development of non-AIDS-related comorbidities, including atherosclerosis, cardiovascular disease, and periodontal disease [16,17].

### 1.2. Effects of HIV Infection on the Gut Microbiota

The majority of studies analyzing stool samples utilize next-generation sequencing techniques, predominantly 16S rRNA sequencing. In the 16S rRNA sequencing process, a specific region of the 16S rRNA gene is first amplified using polymerase chain reaction with primers that target highly conserved sequences, followed by sequencing. However, the literature suggests that this method may lack the sensitivity necessary to detect subtle alterations in the microbiota. Additionally, it does not always directly sequence specific genes; rather, it predicts their presence based on the amplified regions [18]. In contrast, the other two next-generation sequencing methods—shotgun metagenomic sequencing and RNA sequencing—analyze all DNA and RNA present in the sample, respectively. These methods are better suited for species-level analysis [19]. The third primary technique is whole genome sequencing, which utilizes random primers to sequence overlapping segments of a genome. This method is more reliable for species-level identification, although it is significantly more expensive [20]. Studies examining changes in microbiota composition due to HIV infection detail findings related to microbiota composition, including alpha diversity (diversity within a single sample, characterized by evenness and richness) and beta diversity (variation in taxa composition among different samples within a habitat) indices, as well as microbiota and metabolomic composition [1]. Additionally, the literature often describes alterations in systematic markers and bacterial translocation markers in circulation that are linked to changes in the microbiota. In the following sections, we will adhere to this analytical framework.

Although the data in the literature are to some extent conflicting, it is implicated that HIV infection leads to a reduction in alpha diversity, specifically in terms of species richness, compared to seronegative controls. Regarding beta diversity, studies have consistently reported distinct clustering of the microbiota in people living with HIV (PLWH) compared to seronegative controls [21,22,23].

In seronegative individuals, the dominant bacterial populations in the large intestine include *Bifidobacterium*, *Bacteroides*, *Lactobacillus*, *Clostridium*, *Fusobacterium*, and *Enterobacteria* [24]. Due to HIV infection, *Proteobacteria*, *Fusobacteria*, *Bacteroidetes*, and *Firmicutes* phyla change is the most frequently reported, with the phylum *Proteobacteria* showing an enrichment of *Prevotella* and *Enterobacteriaceae*, especially in potential pathogens such as *Klebsiella*, *Succinivibrio*, *Escherichia–Shigella*, *Megasphaera*, and *Ruminococcus gnavus*. Also, multiple data indicate that fecal samples of PLWH showed a decrease of *Bacteroides* and *Clostridia* [25,26,27]. *Prevotella* and *Enterobacterales* have been associated with bacterial translocation and chronic immune activation [28]. However, data suggest that specific *Prevotella* species, such as *Prevotella copri*, exhibit interspecies variations that may result in less pronounced proinflammatory properties [29,30].

*Prevotella* enrichment has also been suggested to be a preliminary alteration observed among men who have sex with men (MSM) [31,32]. In addition to microbiota changes, alterations in lipid, amino acid, and carbohydrate metabolism have also been observed among the MSM population [33]. The literature data also suggest that MSM may not be a major driver for microbiome changes [31].

Also, a subset of PLWH, known as “elite controllers”, can maintain viral load without the need for ART. This population exhibits a gut microbiota and metabolome composition similar to that of seronegative controls [34]. Nascimento et al. examined fecal samples from elite controllers, non-controllers, and seronegative controls, suggesting that the genus *Lachnospiraceae UCG-004* may serve as a potential marker for HIV control [35]. Additionally, Sperk et al. found that the gut microbiome of elite controllers showed significant enrichment of the genus *Prevotella*, along with an elevation of dipeptides that contributed to this enrichment [36].

In terms of the mycobiota, *Debaryomyces hansenii*, *Candida albicans*, *Candida parapsilosis*, *Saccharomyces*, *Malassezia*, *Cladosporium*, and *Aspergillus* have all been identified as the most abundant fungi. Notably, *C. albicans* dysbiosis is strongly linked to the gastrointestinal symptoms of PLWH [37,38,39]. The gut virome of PLWH is characterized by an enrichment of bacteriophages and significant interindividual variability [40]. Research also points to the enrichment of *adenoviruses*, *Adenoviridae*, and *Anelloviridae*, with the latter two showing particularly high levels in people with AIDS [40,41]. Further analysis of the gut microbiota composition is over the limits of the current review; for more detailed information, we would like to refer to additional reviews [26,42,43].

Besides HIV infection, ART [44], treatment status [45], sexual preferences [46], age [47], route of infection [35], and geography [6] are additional factors that contribute to the dysbiosis of the gut microbiome and the development of an altered gut homeostasis state in PLWH. The literature indicates that bacterial biodiversity in the gut increases with age until approximately 40 years, after which this increase levels off [48]. In adults, the gut microbiome predominantly consists of *Firmicutes*, with *Bacteroidetes* being the second most abundant phylum. Changes in the ratio of these phyla are strongly associated with aging [49]. Furthermore, age-related alterations in the microbiome are often accompanied by a biological process characterized by the progressive decline of the immune system, leading to age-associated chronic inflammation and dysregulation of the microbiota [50]. Regarding gender differences, data suggest that women may exhibit greater microbial biodiversity than men. This disparity can be attributed to various factors, including hormonal differences, dietary habits, medication use, body mass index, and colonic transit time. Additionally, components of the immune system possess specific receptors for sex hormones, indicating a potential influence on the symbiotic relationship between the host and the immune system [48,51].

Furthermore, the effects of prebiotics and probiotics have been extensively researched as potential interventions for disease-associated dysbiosis [52,53]. Studies indicate that targeted interventions involving specific microorganisms can alter the complex interactions within the microbiome, leading to health benefits for the host, such as the restoration of dysbiosis to compositions similar to those found in healthy individuals. Additionally, research suggests that these interventions may positively influence immune homeostasis, promoting a less proinflammatory environment [50,54]. However, these studies regarding pre- or probiotic interventions in HIV infection remain inconclusive, with some research yielding mixed results and systematic reviews indicating potential bias or insufficient evidence [53,55,56].

### 1.3. Effects of HIV Infection on the Gut Metabolome

In addition to microbiota alterations, dysbiosis also results in significant changes to the metabolome [5,57]. Notably, members of the anaerobic *Firmicutes* phylum, most notably the *Ruminococcaceae* and *Lachnospiraceae* families, are responsible for producing short-chain fatty acids (SCFAs), which serve as energy sources for enterocytes and play a key role in maintaining the integrity of the gut barrier [58,59,60]. On the other hand, Koay et al. suggested that SCFAs, especially acetate and butyrate, can also reactivate latent HIV infection [61]. While HIV infection has been associated with a relative depletion of the *Firmicutes* phylum, this does not necessarily lead to a decrease in SCFA levels, as certain species within the phylum may increase [5,28,59].

Another critical metabolite affected by dysbiosis in HIV is tryptophan, an essential amino acid involved in the synthesis of proteins, melatonin, and serotonin. Tryptophan metabolism has been implicated in the progression of HIV/AIDS, with disruptions in its availability and processing linked to disease severity [62]. Additionally, HIV-associated wasting disease has been connected to the inability of HIV-associated microbiota to produce essential amino acids such as lysine, proline, and phenylalanine [13].

### 1.4. Effect of HIV Progression and the Effect of AIDS on the Gut Microbiota

Untreated HIV infection progresses to an advanced stage known as AIDS, which is characterized by the onset of AIDS-defining illnesses and a CD4 T cell count below 200 cells/mm^3^ [63]. Research suggests that the gut microbiota plays a significant role in the decline of CD4 T cell counts, contributing to AIDS pathogenesis [8,9,32]. Conversely, AIDS is also associated with alterations in the gut microbiota composition [32,64]. Also, in AIDS, prophylactic antifungal or antiviral therapies can be also warranted, further altering gut composition [65].

Several studies indicate that CD4 cell count, particularly when it falls below 200 cells/mm^3^, is one of the primary factors influencing microbial translocation, systematic immune activation, and microbial richness, consequently creating reduced alpha diversity [32,66]. However, beta diversity, or bacterial composition, has not shown consistent patterns aside from CD4+ T cell depletion when compared to PLWH without AIDS [32,41]. The bidirectional relationship between AIDS and gut microbiota is evident in that HIV progression and the ensuing systemic inflammation are positively correlated with the enrichment of opportunistic bacteria and the depletion of SCFA-producing or anti-inflammatory bacteria [66] Specifically, opportunistic pathogens such as *Erysipelotrichaceae*, *Enterobacteriaceae*, *Desulfovibrionaceae*, and *Fusobacteria* increase in abundance, while beneficial bacteria such as *Lachnospiraceae*, *Ruminococcaceae*, *Bacteroides*, and *Rikenellaceae* decrease [64,67,68]. Additionally, beneficial bacteria such as *Bifidobacterium* and *Lactobacillus*, known for their protective roles, are depleted in people with AIDS [64,68]. Furthermore, *Fusobacterium* is negatively associated with CD4+ T cell counts but positively correlated with CD4+ T cell activation and regulatory T cell levels [65,69]. The enrichment of facultative anaerobes and the reduction of obligate anaerobes further suggest increased intestinal permeability and a decline in SCFA production [70].

### 1.5. Effects of HIV Infection on Systematic Elevation Markers

Multiple studies have also analyzed systematic inflammation marker alterations in HIV infection. Lipopolysaccharides (LPSs), which correlate positively with levels of Gram-negative bacteria, induce local inflammation, thereby contributing to the breakdown of gut–blood barrier integrity. LPS-binding protein (LBP), secreted by enterocytes, either binds to LPS in the bloodstream to neutralize it or activates mononuclear cells [15,71]. Once monocytes are activated, they express CD14 receptors, and the soluble form of this receptor, soluble CD14 receptors (sCD14), can be measured in plasma. Additionally, intestinal fatty acid-binding protein (I-FABP) serves as a marker for intestinal epithelial cell damage, reflecting the degree of gut barrier disruption [15,71].

HIV infection has also been linked with elevated levels of interleukin-6 (IL-6), tumor necrosis factor-alpha (TNF-α), soluble CD14 (sCD14), beta-D-glucan (BDG), and occludin, although comprehensive data are lacking to confirm these findings [72,73,74,75].

### 1.6. Antiretroviral Therapy

According to current guidelines, initiation of ART is warranted immediately upon HIV diagnosis to restore immune function and increase CD4 counts. Currently, most common ART regimes are composed of eight classes of drugs: nucleoside reverse transcriptase inhibitors (NRTIs), non-nucleoside reverse transcriptase inhibitors, protease inhibitors (PIs), integrase strand transfer inhibitors, fusion inhibitors, chemokine receptor antagonists, post-attachment inhibitors, and capsid inhibitors. Each class targets different stages of the HIV replication cycle [76,77]. Standard therapy typically consists of a combination of three active agents, mainly including two NRTIs paired with either an INSTI, a boosted PI, or an NNRTI [78]. Recently updated guidelines prefer INSTIs or NNRTIs as a backbone of the regime for naïve patients [79,80,81]. Therefore, in our study, we focused on comparing these two groups of antiretrovirals most commonly added to NRTIs in clinical practice.

INSTIs target the integrase enzyme in the replication cycle of both HIV-1 and HIV-2. They are widely used in standard antiretroviral therapies due to their favorable tolerability, limited drug–drug interactions, robust efficacy, and high barrier to resistance mutations [79,82]. Evidence is slowly increasing on the side effects of INSTIs, partly on the question of weight gain, although the underlying mechanism remains unclear. Less commonly reported side effects include headaches, fatigue, diarrhea, nausea, and insomnia [78,83,84].

NNRTIs, on the other hand, inhibit HIV replication by blocking cDNA elongation at a site distinct from that targeted by NRTIs, providing a unique antiviral effect against HIV-1 [77,85]. NNRTIs are highly specific and exhibit low toxicity, making them favorable in ART regimes [86]. However, they possess a lower barrier to resistance, leading to a greater risk of genetic mutations compromising their effectiveness [79,87].

Current evidence suggests that either untreated or treated HIV infection is still associated with gut microbiota dysbiosis, increased inflammation, and consequently bacterial translocation. Even after long-term ART treatment and with a suppressed viral count, the bacterial composition and richness remained distinct from seronegative controls [88,89,90]. A key factor contributing to this persistent dysbiosis is the existence of HIV reservoirs within long-lived cells or anatomical sites, where the virus can continue to replicate, thereby sustaining chronic inflammation. The literature data suggest the gut CD4 T cells as an important possible site for the reservoir. These reservoirs maintain resistance to both ART and the immune system and are responsible for viral rebound if ART is discontinued [91,92,93]. Immunological nonresponders who fail to recover adequate CD4 counts, despite viral suppression, are at increased risk for comorbidities and higher mortality [94].

Furthermore, several studies indicate that antiretroviral agents can cause shifts in specific phyla, orders, or species within the gut microbiota. However, the exact mechanisms by which these changes occur, the specific bacteria targeted by different antiretroviral drugs, and how these changes contribute to gut dysbiosis remain unclear. There are, however, several theories. Directly, this may be partially attributed to the antimicrobial properties of certain ART drugs [24]. For instance, a Swedish in vitro study demonstrated that the NRTI zidovudine and the NNRTI efavirenz exhibit antimicrobial activity against specific bacteria [24]. Another potential explanation involves the pharmacokinetic properties of ART drugs and their differing ability to penetrate the intestinal tract, reaching systemic circulation at varying rates and doses [95]. Also, ART agents may have a direct effect at mucosal sites to induce inflammation and increase permeability [96].

Indirectly, another hypothesis suggests that ART may also influence the gut phageome, which plays a role in bacterial network coordination, further altering the microbial environment [97,98]. Moreover, side effects of ART, such as weight gain and increased BMI linked to INSTIs, can independently impact the composition of the microbiota [88]. Furthermore, HIV infection usually shows enrichment of SCFA-producing bacteria, while on ART treatment, patients’ fecal samples were depleted in those bacteria [99].

The role of the gut microbiota in HIV infection has emerged as a significant area of research. However, relatively few studies have examined the effects of ART on the gut microbiome. Current literature is being developed to understand the specific impacts of each ART regime on the microbiota and the resulting alterations. Additionally, researchers are investigating whether these microbiota shifts could indirectly contribute to improved immune reconstitution. Furthermore, ongoing studies are attempting to assess whether one HIV therapy could outperform another in terms of efficacy or could facilitate faster immune recovery. PI-based treatments are receiving less attention due to their side effect profiles and potential drug–drug interactions [79]. Conversely, INSTI-based therapies have gained a more prominent role in treatment guidelines, while NNRTIs remain part of first-line treatment regimens.

### 1.7. Aims of the Scoping Review

Our aim was to conduct a scoping review of the evidence on shifts in the gut microbiota following HIV/AIDS, particularly in relation to the initiation of antiretroviral therapy.

Three key research questions are addressed in this scoping review:Mapping the gut microbiome alterations in PLWH who are receiving INSTI-based therapy compared to ART-naïve PLWH.Mapping the gut microbiome alterations in PLWH who are receiving an NNRTI-based regime compared to ART-naïve PLWH.Mapping the gut microbiome alterations in PLWH who are receiving INSTI-based therapy compared to those on an NNRTI-based regime.

## 2. Materials and Methods

This scoping review followed the guidelines set out by the PRISMA (Preferred Reporting Items for Systematic Reviews and Meta-Analyses) extension for scoping reviews [100]. Systematic literature searches were conducted in PubMed (3 September 2024), Web of Science (3 September 2024) and Embase (16 October 2024) from their inception, without applying any language or publication filters. Both Medical Subject Headings (MeSH) and free-text keywords were used to ensure a thorough search. The search strategy was tested against eight preselected studies known to be relevant and was validated accordingly. The detailed search strategy and the PRISMA extension for scoping reviews checklist can be found in Appendix A respectively. Additionally, reference lists of relevant systematic reviews were manually searched to identify any further eligible studies.

Study selection took place in two stages: (1) screening of titles and abstracts, and (2) full text review, in accordance with the detailed inclusion and exclusion criteria (Table 1). Both stages were performed by the first reviewer, with the second stage being double-checked by the second reviewer. The process was executed using Excel (version 2013). Any potential discrepancies were resolved by discussion or involving a third reviewer.

Relevant data were extracted by one reviewer and checked by another. The following information was extracted: study title, authors, year of publication, country of origin, study design, number of participants, participant characteristics, intervention specifics, methods of data extraction from stool samples, follow-up duration, and key outcomes (including alpha and beta diversity indices, microbiota composition, and alterations in bacterial translocation or systemic inflammation markers).

The study inclusion process is illustrated in Figure 1, while the data from the selected studies are summarized in Table 2, Table 3 and Table 4 to provide a clear overview of the existing research in this area, as well as to identify any potential research gaps. As this is a scoping review, a formal risk of bias assessment was not performed.

## 3. Results

### 3.1. Overview

A total of 11,619 articles were identified from the PubMed, Web of Science, and Embase databases. After removing 3316 duplicates, 8274 records were excluded based on title and abstract screening due to irrelevant topics, non-English language, or unsuitable publication types. Of the 29 remaining reports, additional exclusions were made for reasons such as missing full text, lack of detailed information on ART, or the use of different ART regimes (as outlined in Figure 1). Ultimately, two randomized controlled trials and eight case-control studies were included.

### 3.2. Comparison of Gut Microbiota Composition in INSTI-Treated Patients and ART-Naïve Individuals

Our first research question aimed to map the evidence regarding gut microbial shifts between INSTI-treated and ART-naïve PLWH. In total, we identified two case-control studies addressing this research question.

Villoslada-Blanco et al. examined the microbiota composition in PLWH treated with INSTIs (*n* = 15) and compared them to the gut compositions of ART-naïve PLWH (*n* = 15) and seronegative individuals (*n* = 26). PLWH receiving INSTI-based treatment were administered either second-generation dolutegravir or bictegravir for at least one year. All included patients were immune responders. Stool samples were analyzed using 16S rRNA sequencing [101]. According to the study results, the alpha and beta diversity were reduced in ART-naïve PLWH, whereas INSTI treatment restored alpha diversity. Changes in alpha diversity were significant only when assessed using the Chao1 index. However, no significant statistical difference in beta diversity was observed between ART-treated and ART-naïve PLWH [101]. The study further highlighted that INSTI treatment reduced the elevated abundance of *Prevotella 2* (*Bacteroidetes* phylum) among PLWH, linking this bacterium to Th17-mediated mucosal inflammation [101]. Additionally, compared to seronegative controls, an increase in the *Spirochaetes* and *Cyanobacteria* phyla and a decrease in the *Bacteroidetes* and *Actinobacteria* phyla were observed, suggesting a reduction in inflammation [101]. Regarding bacterial translocation and systemic inflammation, the study results suggested that INSTI treatment significantly reduced levels of biomarkers such as sCD14, LBP, and fecal calprotectin compared to the effects of ART-naïve PLWH [101]. IL-6 levels remained unchanged, while TNF-alpha levels returned to normal with INSTI treatment [101]. SCFA levels were also measured, with both ART-naïve and ART-treated PLWH showing elevated levels compared to seronegative controls [101]. However, ART treatment did not significantly alter SCFA levels in comparison to ART-naïve PLWH [101].

Villoslada-Blanco et al. also investigated the gut virome composition on the same patient population [97]. The study revealed that bacteriophages were the most abundant component of the gut virome, with a restoration of richness following INSTI treatment. Changes in alpha diversity were observed only when using Fisher’s alpha index. Nevertheless, beta diversity changes in the bacteriophage virome were independent of INSTI treatment. Additionally, no significant changes in alpha or beta diversity of eukaryotic viruses were observed between subgroups [97]. The study further emphasized that plant- and fungal-infecting viruses were more abundant than animal-infecting viruses. Despite the significant changes of alpha or beta diversity among subgroups, elevated lysogenic phage levels and reduced *Proteobacteria*-infecting phage levels persisted [97].

The study results are summarized in Table 2.

### 3.3. Comparison of Gut Microbiota Composition in NNRTI-Treated Patients and ART-Naïve Individuals

For our next research question, we aimed to map the available literature comparing the impact of NNRTI-based ART treatments on the gut microbiota with the effects of untreated HIV infection. We identified three case-control studies and one randomized controlled trial.

Sortino et al. conducted a study on recently diagnosed MSM PLWH (*n* = 52) who initiated NNRTI-based regimes, following them for six months [105]. Fecal samples were collected before and after ART initiation and compared to the samples of seronegative MSM controls (*n* = 7) [105]. Study results demonstrated that NNRTI treatment significantly altered the gut microbiome compared to ART-naïve patients and led to an enrichment of proinflammatory bacteria [105]. Additionally, the elevation of *Fusobacterium*, increased levels of sCD14, and polymorphonuclear cell infiltration in the lamina propria of the intestine persisted independently of ART initiation [105]. At the phylum level, *Bacteroidetes* depletion and *Proteobacteria* enrichment were also observed after NNRTI initiation. Regarding diversity indices, alpha diversity remained lower, as measured by the Chao1 and Shannon indices, even after six months of ART treatment. However, beta diversity indices showed partial restoration [105]. In the metabolomic analysis, the study also found that tryptophan, an amino acid reduced in recently diagnosed PLWH, was restored to normal levels following NNRTI treatment. Nonetheless, no bacterial taxa were found to correlate with the tryptophan metabolism pathway [105].

Ray et al. conducted a one-year study on ART-naïve PLWH initiating either NNRTI-based or PI-based conventional regimes [24]. The discussion here is focused exclusively on the evidence presented in the study regarding the effects of NNRTI-based treatment. The study reported that NNRTI-based ART did not restore microbial diversity after one year, with alpha diversity continuing to decline [24]. Alpha diversity indices were measured using the Chao1, Fisher, and ACE indices. Furthermore, in beta diversity analysis, only insignificant clustering changes were observed following NNRTI initiation. Both NRTI and NNRTI exhibited direct antimicrobial activity against *Bacteroides fragilis*, *Prevotella*, and *Enterococcus faecalis* [24]. *Enterococcus* faecalis plays a role in early gut development and inflammation, with probiotic properties [24]. *Bacteroides* and *Prevotella* are anaerobic bacteria known for their ability to produce polysaccharides, which contribute to regulating dysbiosis [24]. In terms of microbiota composition, the study revealed an enrichment of *Firmicutes* and a depletion of *Bacteroidetes* at the phylum level during NNRTI therapy [24]. At the genus level, there was a notable reduction in *Lachnospira*, *Oribacterium*, *Oscillospira*, and *Prevotella* [24]. The authors highlighted that HIV infection particularly affects genera such as *Bifidobacterium*, *Lactobacillus*, *Faecalibacterium*, and *Lachnospira* [24]. Additionally, *Prevotella* and the *Enterobacterales* families were found to be enriched in HIV infection, contributing to proinflammatory properties [24].

Kantamala et al. investigated acutely diagnosed PLWH (*n* = 36) before initiating NNRTI treatment and followed them for 48 weeks [15]. Plasma levels of LBP, sCD14, I-FABP, and circulating gut-homing CD4 and CD17 T cells were measured at baseline and at 24- and 48-week follow-ups [15]. Plasma 16S rDNA levels were also measured as a marker of microbial translocation. No significant differences were observed in 16S rDNA levels after 48 weeks of NNRTI-based therapy. However, LBP levels decreased significantly, independent of plasma 16S rDNA levels [15]. The study reported no significant changes in sCD14 or I-FABP levels. Gut-homing CD4 T cell counts increased, though the increase in Th17 cell counts was insignificant [15]. Based on the results, the study suggested a trend toward immune recovery after a short course of NNRTI. However, 16S rDNA levels showed no correlation with I-FABP, sCD14, or gut-homing T cell levels, leading the study to suggest that short-term ART did not significantly reduce systemic inflammation [15].

Ji et al. examined stool and blood samples from individuals with acutely diagnosed PLWH. The patients were stratified into subgroups based on their baseline CD4 T cell counts (<300/mm^3^ or >300/mm^3^) and followed for 12 months after the initiation of NNRTI therapy [102]. After 12 months, alpha diversity did not differ significantly overall. However, subgroup analysis revealed a significant increase in alpha diversity among those with baseline CD4 T cell counts <300/mm^3^. In terms of beta diversity, a significant change was observed following NNRTI treatment [102]. Also, levels of I-FABP and sCD163 significantly decreased post-NNRTI therapy, although further analysis revealed major restoration only among those with baseline CD4 T cell counts <300/mm^3^. Regarding bacterial composition, following NNRTI treatment, there was a relative increase in the phyla *Proteobacteria* and *Fusobacteria*, along with their subtaxonomies, while the *Bacteroidetes* phylum showed a relative decrease [102]. Within the *Firmicutes* phylum, the *Ruminococcaceae* family and the *Faecalibacterium* genus decreased. In subgroup analysis, among individuals with baseline CD4 T cell counts >300/mm^3^, the difference in the order *Fusobacteriales* became insignificant, and among those with CD4 T cell counts <300/mm^3^, the elevation in the order *Bacillales* was no longer significant [102].

The study results are summarized in Table 3.

### 3.4. INSTI-Based Treatment Regime Compared to NNRTI-Based Treatment

Our next research question focused on mapping the effects of INSTI-based and NNRTI-based treatments on the gut microbiota. For the research question, three case-control studies and one randomized controlled trial were identified.

Narayanan et al. investigated microbiota differences between PLWH (*n* = 69) treated with INSTI-based (*n* = 54) and NNRTI-based (*n* = 13) regimes and compared them to seronegative controls (*n* = 80) [25]. The study also examined the correlations between ART and body mass index (BMI) [25]. According to the results, INSTI-based therapy was associated with an enrichment of *Faecalibacterium* and *Bifidobacterium*, while NNRTI-based treatment led to an enrichment of *Gordonibacter*, *Megasphaera*, and *Staphylococcus* [25]. Further analysis regarding the specific second generation INSTI treatments reported *Bifidobacterium*, *Butyricimonas*, and *Butyricicoccus* enrichment in the bictegravir subgroup and *Faecalibacterium* and *Ruminococcus gauvreauii* elevation among PLWH on dolutegravir treatment. The study also suggested that the enrichment of *Faecalibacterium* and *Bifidobacterium* during INSTI therapy may indicate more advanced immune reconstitution compared to the NNRTI regime [25]. Conversely, NNRTI-treated PLWH exhibited an enrichment in *Megasphaera*, a bacterium known for producing short-chain fatty acids, which serve as an energy reserve in the gut [25]. Notably, fecal samples from PLWH with a high BMI showed elevated levels of *Bifidobacterium* and *Dorea* [25].

Villanueva-Millán et al. conducted a cross-sectional study comparing the effects of INSTI- (*n* = 8), PI- (*n* = 15), and NNRTI-based (*n* = 22) regimes on the gut microbiota of PLWH (*n* = 45) with those of untreated PLWH (*n* = 5) and seronegative controls (*n* = 21) [44]. We focused on mapping the effects of NNRTI- and INSTI-based treatments in the study findings. Analysis of fecal samples revealed that the INSTI-based regime induced the greatest increase in alpha diversity, closely resembling the levels found in seronegative controls, while the NNRTI-based regime achieved only partial recovery of diversity [44] Alpha diversity analysis was conducted using the number of species, as well as the Alpha, Margalef’s diversity, and Chao1 indices, all of which demonstrated significant associations. Furthermore, the study reported the most notable alterations regarding bacterial composition at the species level, particularly within the *Firmicutes* phylum and *Clostridiales* class [44]. INSTI-based therapy was reported to be associated with a smaller decline in the *Clostridiales* class and reduced abundance of the *Lachnospiraceae* families [44]. Additionally, a specific reduction in *Desulfovibrio* sp., a bacterium from the *Desulfovibrio* genus known for producing hydrogen sulfide, a recognized cytotoxic agent, was observed during INSTI treatment [44]. Conversely, NNRTI-based therapy led to an increase in the *Lachnospiraceae* families and the *Pseudomonas* genus, while it caused a reduction in the *Bacteroidales* order, the *Bacteroidaceae* family, and the *Streptococcus* genus [44]. In terms of systemic inflammation, the levels of sCD14 and IL-6 in ART-treated PLWH remained comparable to those of seronegative controls [44].

Hanttu et al. conducted a study on PLWH who had previously been treated with NNRTI- or PI-based ART regimes (*n* = 41) and were switched to an INSTI-based regime (*n* = 19), with follow-up over 24 weeks during which fecal samples were collected [103]. The study compared these PLWH to those who remained on their original ART regimes (*n* = 22) and to seronegative controls (*n* = 10) [103]. After 24 weeks, patients on INSTI-based ART exhibited alpha diversity levels that closely approximated those of seronegative controls and were significantly higher than those observed in the other treatment regimes [103]. The switch from NNRTI to INSTI treatment resulted in an elevation of *Prevotella*, alongside reductions in *Phascolarctobacterium* and *Bacteroides* [103]. The study also assessed bacterial translocation markers, including I-FABP and serum LBP, which remained unchanged throughout the 24-week follow-up [103].

Fu et al. examined the gut microbiota of MSM patients and compared acutely diagnosed ART-naïve PLWH (*n* = 30) to those on NNRTI-, INSTI-, or PI-based treatments (*n* = 30), as well as to seronegative non-MSM controls (*n* = 30) [104]. Among ART-treated PLWH, those on NNRTI regimes displayed the lowest alpha diversity, with notable reductions in richness indices, while evenness remained largely unchanged [104]. Additionally, NNRTI therapy was reported to be associated with reduced beta diversity [104]. In contrast, study results regarding an INSTI-based regime did not show similar alterations [104]. At the phylum level, NNRTI therapy increased *Fusobacteria* while decreasing *Actinobacteria* and *Euryarchaeota* [104]. Genus-level changes included *Fusobacterium* enrichment and reductions in *Faecalibacterium* and *Escherichia* [104]. The study found a strong negative correlation between *Fusobacterium* abundance and alpha diversity [104]. The study also noted that INSTI treatment, in contrast, resulted in diversity and inflammation levels more closely resembling those of seronegative controls [104].

The study results are summarized in Table 4.

## 4. Discussion

In the present study, our aim was to map the impact of different ART regimes on gut microbiota diversity and composition, in comparison to the effects of HIV infection in the absence of ART. Evidence from the included studies indicates that INSTI-based regimes typically enhance alpha diversity, approximating the levels observed in seronegative controls, while beta diversity remains relatively unchanged. The results also suggest that INSTI therapies improve microbiota composition and may help reduce inflammatory markers. In contrast, NNRTI-based treatments show only limited improvements in alpha diversity and are associated with an increase in proinflammatory bacteria.

In detail, we begin by summarizing existing knowledge on changes in alpha and beta diversity indices following the initiation of INSTI or NNRTI treatment. Alpha diversity measures were assessed in the collected studies using multiple methods, including Chao1, Shannon index, number of species, Alpha, Margalef’s diversity, and Fisher’s alpha indices, each measuring different aspects of alpha diversity [106]. Although not consistently observed with all indices, the analysis of the effects of INSTI on the gut microbiota suggested a restoration of alpha diversity indices [44,101,103]. In contrast, analyses of NNRTI-based regimens suggested a sustained reduction in diversity, with the exception of findings by Villanueva-Millán et al., who reported partial restoration and Ji et al., who reported restoration only in PLWH with baseline CD4 T cell counts < 300/mm^3^ and a non-significant decrease in those with CD4 T cell counts > 300/mm^3^ [24,44,102,104,105]. Villanueva-Millán et al. also suggested that the more pronounced restoration of alpha diversity in INSTI-treated PLWH may be attributable to the superior reduction of proviral DNA, owing to the pharmacological dynamics of INSTIs [44].

Regarding beta diversity, the included studies indicated that neither INSTI nor NNRTI treatments significantly affected beta diversity, with the exception of Fu et al. and Ji et al., who reported notable clustering alterations in beta diversity between NNRTI-treated PLWH and ART-naïve individuals [24,101,102,104,105]. Ji et al. suggested that the differences in their alpha and beta diversity results compared to previous studies might be attributable to the lack of accounting for the confounding effects of immune status in PLWH [102].

Studies have also examined the alterations in microbiota composition. Villoslada-Blanco et al. suggested that INSTI treatment led to a reduction in *Bacteroidetes* (specifically *Prevotella 2*) and *Actinobacteria*, alongside an increase in the phyla *Spirochaetes* and *Cyanobacteria*, thus shifting the microbiota composition in a manner that may attenuate inflammation. Notably, SCFA levels were found to be independent of INSTI therapy [101]. Furthermore, Narayanan et al. reported an enrichment of *Faecalibacterium* and *Bifidobacterium* following INSTI treatment, whereas NNRTI treatment was associated with an enrichment of *Gordonibacter*, *Megasphaera*, and *Staphylococcus*. The study summarizing these findings suggested that the enrichment of both *Faecalibacterium* and *Bifidobacterium* during INSTI treatment may indicate a more advanced state of immune reconstitution compared to NNRTI treatment. Conversely, it also reported an elevation of *Megasphaera* during NNRTI treatment, which is recognized as a SCFA producer in the gut [25]. Moreover, Hanttu et al. investigated the gut microbial composition of PLWH who transitioned from NNRTI to INSTI therapy and observed an enrichment of *Prevotella*, along with reductions in *Phascolarctobacterium* and *Bacteroides*. Their study suggested that *Bacteroides* species are linked to the promotion of regulatory T cells, while *Phascolarctobacterium* is noted for its role in SCFA production [103]. The findings of Villoslada-Blanco et al. and Hanttu et al. also indicated that *Prevotella* may exhibit interspecies variations [101,103]. Additionally, Villanueva-Millán et al. identified a reduction in SCFA-producing bacteria [44].

The effects of NNRTI treatment have also been explored. Fu et al. found an increase in *Fusobacteria* during therapy, which correlated negatively with alpha diversity, suggesting that *Fusobacteria* may be a primary driver of gut dysbiosis [104]. In contrast to the findings of Narayanan et al., Ray et al. observed a decrease in *Lachnospira*, *Oribacterium*, and *Oscillospira* during NNRTI treatment, all of which are known SCFA producers [24]. Furthermore, Sortino et al. indicated a persistent proinflammatory environment, characterized by an elevation of *Fusobacterium*, *Proteobacteria*, and a depletion of *Bacteroidetes* during NNRTI treatment, although neither systemic markers of inflammation nor alterations in the tryptophan pathway were significantly associated with either subgroup of microbiota [105].

Regarding virome composition, Villoslada-Blanco et al. identified a dominance of bacteriophages alongside lower levels of eukaryotic viruses. In contrast to the findings of Monaco et al., this study indicated that plant- and fungal-infecting viruses were more prevalent than animal-infecting viruses. This difference may be stemmed from Monaco et al.’s exclusive focus on DNA viruses. Furthermore, Villoslada-Blanco et al. highlighted a reduction in *Proteobacteria*-infecting phages [97]. These findings aligned with changes observed in the bacteriome, particularly the elevation of *Proteobacteria* and *Succinivibrio*, the latter of which is associated with symptoms such as abdominal pain and diarrhea [83,84].

In terms of mapping evidence on systemic inflammatory markers, Villoslada-Blanco et al. suggested that INSTI treatment resulted in reductions of inflammatory markers (sCD14, LBP, and fecal calprotectin), while Villanueva-Millán et al. and Hanttu et al. reported only a tendency toward reduction [44,101,103]. Regarding NNRTI-based treatment strategies, Villanueva-Millán et al. suggested significant elevations in sCD14 following treatment, while Sortino et al. indicated a sustained proinflammatory environment, characterized by persistent polymorphonuclear cell infiltration in the lamina propria of the intestine and a positive correlation between *Fusobacterium* elevation and sCD14 levels [44,105]. Additionally, Kantamalaa et al. suggested a trend toward immune recovery alongside a sustained inflammatory response during NNRTI treatment [15]. Furthermore, Ji et al. detected a more robust decrease in systemic inflammatory markers among PLWH with baseline CD4 T cell counts <300/mm^3^, while observing only minor changes in those with CD4 T cell counts >300/mm^3^, attributing these limited changes to the direct counteracting effects of ART [102].

A similar review to ours was conducted by Pinto-Cardoso et al. in 2018, evaluating the effects of NNRTI-, PI-, and INSTI-based ART therapies on the gut microbiome [28]. The review included three studies, which reported similar findings regarding alpha and beta diversity [28]. INSTI-based treatment was identified as the most effective in promoting gut microbiota restoration, while PI-based treatment ranked lowest [28]. One key aspect highlighted in the review was the critical importance of accounting for confounding factors in microbiome analysis to ensure accurate results [28].

We believe that our scoping review is the most recent comprehensive assessment of the evidence regarding the effects of INSTI and NNRTI on the gut microbiome since the work of Pinto-Cardoso et al., with additional evidence accumulating in the interim. However, our scoping review has several limitations. First, most microbiome studies included in the review relied on the 16S rRNA sequencing method to determine microbial composition. Furthermore, many studies did not adequately control for confounding factors related to the host, such as baseline microbiota composition and dietary habits, nor for non-host factors, such as geographical distribution, which complicated the homogenization of the studies. Additionally, the included studies recruited a small number of patients for comparison.

## 5. Conclusions

Existing evidence suggests that the gut microbiota plays a role in the chronic inflammation associated with HIV infection, which contributes to the pathogenesis of AIDS, non-AIDS-related comorbidities, and mortality among individuals living with HIV. This chronic immune activation persists independently of ART and is characterized by an altered microbiome, even during long-term viral suppression. While current evidence remains limited and inconclusive, it hints that INSTI-based regimes may support a more effective recovery of the gut microbiome, potentially facilitating enhanced immune reconstitution compared to NNRTI-based treatments.

Further research is needed to clarify the specific effects of different ART regimes on the gut microbiota and to explore the role of the microbiota in disease progression and inflammation, which could guide clinicians in selecting treatments that mitigate gut dysbiosis-induced chronic inflammation.

## Figures and Tables

**Figure 1 microorganisms-12-02221-f001:**
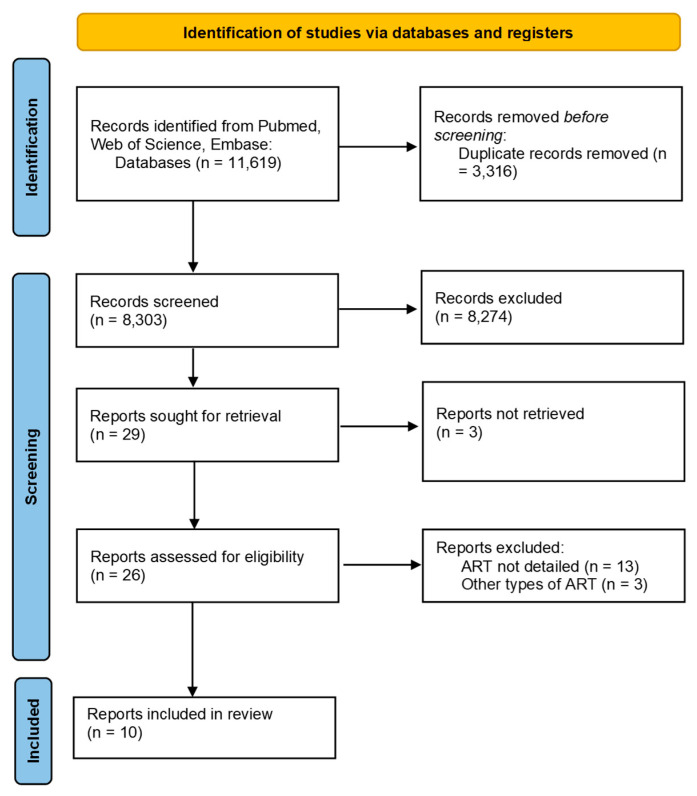
PRISMA flowchart on the study selection process.

**Table 1 microorganisms-12-02221-t001:** Inclusion and exclusion criteria.

**Inclusion Criteria:**			
Studies written in English were considered if they fell into the following categories: reviews, randomized controlled trials, single-arm trials, cohort studies, and case-control studies.			
	**Population**	**Intervention**	**Comparator**
**PICO 1**	Adults diagnosed with HIV infection confirmed by serological tests	PLWH receiving INSTI-based therapy	ART-naïve PLWH
**PICO 2**	Adults diagnosed with HIV infection confirmed by serological tests	PLWH receiving NNRTI-based therapy	ART-naïve PLWH
**PICO 3**	Adults diagnosed with HIV infection confirmed by serological tests	PLWH receiving INSTI-based therapy	PLWH receiving NNRTI-based therapy
**Exclusion criteria:**			
The following types of publications were excluded: case series, case reports, clinical guidelines, non-peer-reviewed literature, conference abstracts, letters, and editorials.			

**Table 2 microorganisms-12-02221-t002:** Literature data on the effect of INSTIs on gut microbiota in PLWH.

Study Title	Author	Publication Year	Participants	INSTI Mediated Alpha Diversity Changes	INSTI Mediated Beta Diversity Changes	INSTI Mediated Changes in Microbiome Composition	Stool Sample Analysis	INSTI Mediated Change on Bacterial Translocation or Systematic Inflammation Markers
**Integrase Inhibitors Partially Restore Bacterial Translocation, Inflammation and Gut Permeability Induced by HIV Infection: Impact on Gut Microbiota**	Villoslada-Blanco et al. [101]	2022	PLWH on INSTI treatment vs. ART-naïve PLWH vs. seronegative controls	INSTI restored alpha diversity (Chao1)	Seronegative controls differed significantly independent of ART treatment	➢ Restored *Prevotella 2* ➢ *Bacteroidetes* phylum ↓ ➢ *Spirochaetes* and *Cyanobacteria* phyla ↑ ➢ *Bacteroidetes* and *Actinobacteria* phyla ↓	16S rRNS gene sequencing	➢ sCD14 ↓ ➢ LBP ↓ ➢ Fecal calprotectin ↓ ➢ IL-6 unchanged ➢ TNF-alpha ↑
**Impact of HIV infection and integrase strand transfer inhibitors-based treatment on the gut virome**	Villoslada-Blanco et al. [97]	2022	PLWH on INSTI treatment vs. ART-naïve PLWH vs. seronegative controls	INSTI restored alpha diversity among bacteriophages (*Fisher’s alpha* indexes)	Seronegative controls differed significantly independent of ART treatment regarding bacteriophage composition	➢ Lysogenic phage ↑ independent of ART ➢ *Proteobacteria*-infecting phage ↓	16S rRNS gene sequencing	-

**Table 3 microorganisms-12-02221-t003:** Literature data on the NNRTI effects on gut microbiota in PLWH.

Study Title	Author	Publication Year	Participants	NNRTI Mediated Alpha Diversity Changes	NNRTI Mediated Beta Diversity Changes	NNRTI Mediated Changes in Microbiome Composition	Stool Sample Analysis	NNRTI Mediated Change on Bacterial Translocation or Systematic Inflammation Markers
**Impact of Acute HIV Infection and Early Antiretroviral Therapy on the Human Gut Microbiome**	Sortino et al. [52]	2020	PLWH before and 6 months after NNRTI treatment vs. seronegative controls	Compared to seronegative controls, alpha diversity was still significantly lower (Chao1, Shannon index)	NNRTI PLWH showed partial restoration	➢ *Rikenellaceae* ↓ ➢ Proinflammatory bacteria ↑ ➢ *Enterobacteriaceae* ↑ ➢ *Bacteroidales* ↓ ➢ *Bacteroidetes* ↓ ➢ *Proteobacteria* ↑ ➢ *Fusobacteria* ↑	16S rRNS gene sequencing	➢ sCD14 ↓ ➢ Fusobacterium was positively associated with sCD14 ➢ Polymorphonuclear cell enrichment did not differ after NNRTI ➢ Tryptophan levels were restored
**Altered Gut Microbiome under Antiretroviral Therapy: Impact of Efavirenz and Zidovudine**	Ray et al. [27]	2021	PLWH before and 10 months after NNRTI treatment vs. seronegative controls	Decreased after NNRTI treatment (Fischer, Chao1, ACE)	Only moderate differences could be observed	➢ *Firmicutes* ↑ ➢ *Bacteriodetes* ↓ ➢ *Prevotellaceae* ↓ ➢ *Lachnospira* ↓, *Oribacterium* ↓, *Oscillospira* ↓, *Prevotella* ↓	16S rRNS gene sequencing	-
**High microbial translocation limits gut immune recovery during short-term HAART in the area with high prevalence of foodborne infection**	Kantamala et al. [15]	2020	PLWH before and 48 weeks after NNRTI treatment	-	-	-	16S rRNS gene sequencing	➢ Plasma 16S rDNA unchanged ➢ LBP ↓ ➢ Gut-homing CD4+ T cell ↑ ➢ Th17 cell unchanged ➢ sCD14 unchanged ➢ I-FABP unchanged
**Changes in intestinal microbiota in HIV-1-infected subjects following cART initiation: influence of CD4+ T cell count**	Ji et al. [102]	2018	PLWH before and 12 months after NNRTI treatment	PLWH with a baseline of <300/mm^3^ CD4 T cells had a significant elevation after NNRTI treatment	Significant differences after NNRTI treatment	➢ *Proteobacteria* ↑ ➢ *Fusobacteria* ↑ ➢ *Bacteroidetes* ↓ ➢ *Ruminococcaceae* family ↓ ➢ *Faecalibacterium* genus ↓	16S rRNS gene sequencing	➢ I-FABP decreased ➢ sCD163 decreased ➢ Major alteration only among PLWH with a baseline of <300/mm^3^ CD4

**Table 4 microorganisms-12-02221-t004:** Study results on the INSTI- versus NNRTI-based treatment effects on gut microbiota in PLWH.

Study Title	Author	Publication Year	Participants	ART Mediated Alpha Diversity Changes	ART Mediated Beta Diversity Changes	ART Mediated Changes in Microbiome Composition	Stool Sample Analysis	ART Mediated Change on Bacterial Translocation or Systematic Inflammation Markers
**Exploring the interplay between antiretroviral therapy and the gut-oral microbiome axis in people living with HIV**	Narayanan et al. [25]	2024	NNRTI vs. INSTI treated PLWH vs. seronegative individuals	-	-	➢ INSTI: *Faecalibacterium* and *Bifidobacterium* ↑ ➢ Bictegravir: *Bifidobacterium*, *Butyricimonas* and *Butyricicoccus* ↑ ➢ Dolutegravir: *Faecalibacterium* and *Ruminococcus gauvreauii* ↑ ➢ NNRTI: *Gordonibacter*, *Megasphaera*, *Staphylococcus* ↑	16S rRNS gene sequencing	-
**Differential effects of antiretrovirals on microbial translocation and gut microbiota composition of HIV-infected patients**	Villanueva-Millán et al. [44]	2017	NNRTI vs. INSTI treated PLWH vs. ART-naïve PLWH vs. seronegative individuals	➢ INSTI: alpha restore ➢ NNRTI: partial restoration	-	INSTI: ➢ δ-*Proteobacteria* ↑ ➢ Less reduction in *Clostridiales* class ➢ Less abundance of *Desulfovibrio* sp. 6 ➢ *Lachnospiraceae* ↓ ➢ NNRTI: ➢ *Lachnospiraceae* families and *Pseudomononas* genus ↑ ➢ *Bacteroidales* order, *Bacteroidaceae* family and *Streptococcus* genus ↓	16S rDNA pyrosequencing	➢ INSTI: sCD14 levels tendency to seronegative controls ➢ NNRTI: sCD14 levels significantly elevated
**Gut microbiota alterations after switching from a protease inhibitor or efavirenz to raltegravir in a randomized, controlled study**	Hanttu et al. [103]	2023	NNRTI-based to INSTI-based vs. PI-based to INSTI-based vs. NNRT-based vs. PI-based ART vs. seronegative controls	INSTI: alpha diversity restored	-	NNRTI to INSTI: ➢ Prevotella 9 ↑ ➢ Phascolarctobacterium ↓ ➢ Bacteroides ↓	16S rRNS gene sequencing	NNRTI to INSTI: I-FABP and LBP unchanged
**Characterization of the intestinal microbiota in MSM with HIV infection**	Fu et al. [104]	2024	NNRTI-based vs. INSTI-based vs. PI-based ART vs. ART-naïve PLWH vs. seronegative controls	NNRTI: reduced alpha diversity compared to ART-naïve PLWH	NNRTI: significantly different beta diversity compared to ART-naïve PLWH	NNRTI: ➢ Fusobacteria ↑ ➢ Actinobacteria ↓ ➢ Euryarchaeota ↓ ➢ Fusobacterium genus ↑ ➢ Faecalibacterium ↓ ➢ Escherichia ↓	16S rRNS gene sequencing	-

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
