# Peer review of "Gut Microbiome Alteration in HIV/AIDS and the Role of Antiretroviral Therapy—A Scoping Review"

_microorganisms, 2024, doi:10.3390/microorganisms12112221_

Reviewer 1 Report
Comments and Suggestions for Authors
1. Following Joanna Briggs Institute (JBI)’s explanation of the research purpose of scoping review, the research content of this paper is not consistent with that of a scoping review. The research questions should align more clearly with the purpose of a scoping review, which is to map the existing literature and identify gaps. Currently, the questions seem overly focused on comparing two specific types of antiretroviral therapy (INSTI and NNRTI) and drawing conclusions, which may be more appropriate for a systematic review or a meta-analysis.
2. The introduction section lacks a clear theoretical underpinning and rationale for why comparing INSTI and NNRTI therapies is particularly important for understanding gut microbiome changes in HIV/AIDS patients. This should be more clearly elucidated.
3. The search strategy is not well-detailed in the methods section. Please provide the full electronic search strategy used to identify studies, including all search terms and limits for at least one database in the main manuscript. The rest should be provided in the Supplementary Material.
4. For a scoping review, it is highly restrictive to rely on only two databases (PubMed and Web of Science) for its search strategy. Many relevant studies might not be indexed in these two databases alone. Please consider searching other databases, e.g. Embase, CINAHL, Scopus, etc.
5. More details on the data charting process are necessary. The data extraction process also seems to lack transparency. Only one reviewer conducted both screening and data extraction, which raises concerns about potential bias. Standard practice involves multiple (independent) reviewers to ensure reliability.
6. In Figure 1, why was "other types of ART (n=2)" excluded? Given the overall small number of studies reviewed, I think it would be more beneficial to include the additional two studies.
7. The discussion of alpha and beta diversity results lacks depth. The authors frequently mention that INSTI-based therapies improve alpha diversity while NNRTI-based therapies show minimal changes. However, no formal risk of bias assessment was conducted, and the validity of the conclusions drawn from the reviewed studies should have been critiqued.
8. The 16S rRNA sequencing method, which is used by most gut microbiota studies, may not be sensitive enough to detect small microbiota alterations (citation: pubmed.ncbi.nlm.nih.gov/37110143). This should be mentioned.
9. Compared to a classic review the number of documents cited is very small, I would therefore expect a greater depth and an analysis of what has been found in common in several articles and what has not. A precise analysis should be added in this section, possibly grouping the screened articles in convenient ways (e.g. grouping by sample type), in order to add useful information for the readers (e.g. for metagenomics studies a comparison in terms of taxa might be relevant).
10. I would caution against making broad conclusions regarding the superiority of INSTI-based therapies based on gut microbiota outcomes. There are several limitations of the evidence, including the fact that several studies used small sample sizes or did not account for confounders like diet, geography, or baseline microbiota composition, etc.
Comments on the Quality of English LanguageThere are multiple grammatical and stylistic issues.
Author Response
Comment 1: Following Joanna Briggs Institute (JBI)’s explanation of the research purpose of scoping review, the research content of this paper is not consistent with that of a scoping review. The research questions should align more clearly with the purpose of a scoping review, which is to map the existing literature and identify gaps. Currently, the questions seem overly focused on comparing two specific types of antiretroviral therapy (INSTI and NNRTI) and drawing conclusions, which may be more appropriate for a systematic review or a meta-analysis.
Response 1: We greatly appreciate your comments on our scoping review. Our intention was to map the evidence regarding the effects of INSTI- or NNRTI-based ART regimens. We have updated the manuscript, accordingly, ensuring that the scoping review guidelines are followed throughout.
Comment 2: The introduction section lacks a clear theoretical underpinning and rationale for why comparing INSTI and NNRTI therapies is particularly important for understanding gut microbiome changes in HIV/AIDS patients. This should be more clearly elucidated.
Response 2: Thank you for this important remark. We have updated the “Antiretroviral therapy” section accordingly
Comment 3: The search strategy is not well-detailed in the methods section. Please provide the full electronic search strategy used to identify studies, including all search terms and limits for at least one database in the main manuscript. The rest should be provided in the Supplementary Material.
Response 3: Thank you for your valuable suggestion. However, we have already included the detailed search strategies for all databases in the Supplementary Material, with their location clearly referenced in the 'Materials and Methods' section of the manuscript. Therefore, we believe that this provides sufficient transparency and prefer to retain the current format without incorporating the search strategy within the main text.
Comment 4: For a scoping review, it is highly restrictive to rely on only two databases (PubMed and Web of Science) for its search strategy. Many relevant studies might not be indexed in these two databases alone. Please consider searching other databases, e.g. Embase, CINAHL, Scopus, etc.
Response 4: We appreciate this comment and have expanded the search strategy to include the Embase research database. The details of the database search have been incorporated into Figure 1 and section ‘3.1. Overview‘.
Comment 5: More details on the data charting process are necessary. The data extraction process also seems to lack transparency. Only one reviewer conducted both screening and data extraction, which raises concerns about potential bias. Standard practice involves multiple (independent) reviewers to ensure reliability.
Response 5: Thank you for your valuable comment. To enhance the robustness of our study, we have invited an additional co-author, Bálint Gergely Szabó, to verify the data extracted by the first reviewer and to improve the clarity of our results. Study selection took place in two stages: 1) screening of titles and abstracts, and 2) full text review, in accordance with the detailed inclusion and exclusion criteria (Table 1). Both stages were performed by the first reviewer, with the second stage being double-checked by the second reviewer. Any discrepancies between the two reviewers were resolved through discussions and by involving the corresponding author. The first reviewer conducted a comprehensive search for relevant studies on PubMed, Web of Science and Embase, extracting data from the selected studies. The included details were study title, authors, year of publication, country of origin, study design, number of participants, participant characteristics, intervention specifics, methods of data extraction from stool samples, follow-up duration, and key outcomes (including alpha and beta diversity, microbiota composition, and alterations in bacterial translocation or systemic markers). The inclusion process is illustrated in Figure 1, while the data from the selected studies are summarized in Tables 2-4. We have also incorporated this information into the ‘Materials and Methods’ section to enhance clarity.
Comment 6: In Figure 1, why was "other types of ART (n=2)" excluded? Given the overall small number of studies reviewed, I think it would be more beneficial to include the additional two studies.
Response 6: Thank you for this important remark. We aimed to map the effects of NNRTI and INSTI-based ART on the gut microbiome exclusively. We chose to omit the two studies that focused on other ART treatment effects for a specific reason. Notably, the different types of ART regimens vary in efficacy, clinical indications for patients living with HIV, side effect profiles, and other factors; but clinical implications of antiretrovirals changed a lot in the past decade restricted to two main preferred groups of antiretrovirals for naïve patients. Consequently, we aimed to provide a clear mapping of evidence of NNRTI and INSTI effects, which are currently by far the most dominantly used groups of antiretrovirals among patients living with HIV.
Comment 7: The discussion of alpha and beta diversity results lacks depth. The authors frequently mention that INSTI-based therapies improve alpha diversity while NNRTI-based therapies show minimal changes. However, no formal risk of bias assessment was conducted, and the validity of the conclusions drawn from the reviewed studies should have been critiqued.
Response 7: Thank you for your comment. Further discussion on alpha and beta diversity has been incorporated into the review at multiple sections. As this is a scoping review with the aim of mapping the evidence, a formal risk of bias assessment was not conducted. Additionally, since a bias assessment was not required, we did not deem it necessary to critique the validity of the conclusions.
Comment 8: The 16S rRNA sequencing method, which is used by most gut microbiota studies, may not be sensitive enough to detect small microbiota alterations (citation: pubmed.ncbi.nlm.nih.gov/37110143). This should be mentioned.
Response 8: Thank you for your valuable suggestion. We have incorporated this into the manuscript.
Comment 9: Compared to a classic review the number of documents cited is very small, I would therefore expect a greater depth and an analysis of what has been found in common in several articles and what has not. A precise analysis should be added in this section, possibly grouping the screened articles in convenient ways (e.g. grouping by sample type), in order to add useful information for the readers (e.g. for metagenomics studies a comparison in terms of taxa might be relevant).
Response 9: Thank you for this important remark. We have updated the “Results” and “Discussion” sections accordingly.
Comment 10: I would caution against making broad conclusions regarding the superiority of INSTI-based therapies based on gut microbiota outcomes. There are several limitations of the evidence, including the fact that several studies used small sample sizes or did not account for confounders like diet, geography, or baseline microbiota composition, etc.
Response 10: Thank you for your valuable suggestions. We have updated the limitations and the "Conclusions" section accordingly.
Reviewer 2 Report
Comments and Suggestions for Authors
This is a timely review of HADs amidst increasing associated factors of environmental contribution and causal/association effects of other NCDs, CDs, and IDs in the pathways utilized.
Areas that could have been further addressed are:clear effect of sex and age which is linked to hormonal differences in the sexes; the mutations common to both inhibitors or clearly unique to a particular inhibitor.
Contribution of WGS, NGS cannot be emphasized enough but not enough mention was made; Further, the significance of "Elite Controllers" could have been given more flesh in relation to bacterial, fungal, and parasitic metabolite dysbiosis.
Finally, more should have been mentioned in regards to diet-prebiotic and probiotic.
Author Response
Comment 1: Areas that could have been further addressed are:clear effect of sex and age which is linked to hormonal differences in the sexes;
Response 1: Thank you for your suggestion. Further information has been added to Section ‘1.2 Effects of HIV Infection on the Gut Microbiota’.
Comment 2: the mutations common to both inhibitors or clearly unique to a particular inhibitor.
Response 2: Thank you for your comment, which is valuable for us for further research, however no useful data was identified by us to incorporate in this specific review. On the other hand given the complexity of the polymorphism and mutations when assessing more groups of antiretrovirals would definitely exceed the dimension of this review. Nevertheless, we are preparing a systematic review specifically on the effect of second generation integrase inhibitors on gut microbiome, where the aspect of mutations and its potential consequences will be integrated.
Comment 3: Contribution of WGS, NGS cannot be emphasized enough but not enough mention was made;
Response 3: Thank you for your insightful comment. An additional discussion has been incorporated into Section 1.2, titled ‘Effects of HIV Infection on the Gut Microbiota’.
Comment 4: Further, the significance of "Elite Controllers" could have been given more flesh in relation to bacterial, fungal, and parasitic metabolite dysbiosis.
Response 4: Thank you for your recommendation. A further discussion has been included in Section „1.2 Effects of HIV Infection on the Gut Microbiota”.
Comment 5: Finally, more should have been mentioned in regards to diet-prebiotic and probiotic.
Response 5: Thank you for your comment. Further information has been added to Section „1.2 Effects of HIV Infection on the Gut Microbiota”.
Reviewer 3 Report
Comments and Suggestions for Authors
The review is based on current studies of gut microbiome alteration in people living with HIV and the effect of antiretroviral drug, especially INSTI and NNRTI on microbiome diversity. By summarizing the study published on PubMed and Web of Science, the review provides evidence that INSTI based ART treatment facilitates better recovery of the gut microbiome. In contrast, NNRTI-based treatments demonstrate limited recovery of alpha diversity, exhibit beta diversity patterns distinct from those of healthy participants and are linked to an increase in proinflammatory bacteria. The review has brought new insights into the current knowledge of gut microbiota and the relation with HIV infection and treatment. I found the paper is interesting to the reader and felt confident that the authors performed careful and professional in the field. However, I have major concerns about the HIV language used in the manuscript and the inaccurate citations during writing. I have come up with several questions and made the comments below to help improve the quality of this review.
Major comments:
1. It should be noted that the National Institute of Allergy and Infectious Diseases (NIAID) has updated HIV language guide to eliminate the use of stigmatizing terminology and advance the use of person-first, inclusive, and respectful language. Some of the more commonly used yet most critical terms to avoid are listed as below, I would suggest the authors read the language guide and change the words used in their manuscript: for example, HIV-infected patient (Line 11, 59, 75, 88), HIV-infected individuals (Line 14, 58, 66, 88). I didn’t list them all, please check carefully throughout the manuscript and change it accordingly.
Stigmatizing Terms To Avoid |
Use These Alternatives |
HIV-infected, HIV-infection*, HIV-positive [people, individuals, populations] |
People living with HIV, people with HIV (*see page 8 for comments on use of “HIV-infection”) |
Subject |
Participant, volunteer |
Sterilizing cure |
HIV eradication, HIV clearance |
AIDS (when referring to the virus, HIV) |
HIV, HIV and AIDS when referring to both |
Mother-to-child transmission |
Perinatal transmission |
Verticals |
Lifetime survivors |
At-risk or high-risk person/population |
Person/population with greater likelihood of ..., high incidence population, affected community |
Target population |
Key population/engage or prioritize a population |
Hard-to-reach population |
Under-resourced, underserved by [specific resource/service], population(s) experiencing discrimination/racism/transphobia |
2. The citations in the manuscript need to be improved significantly. The issues such as not appropriate citation, not citing the original work, and in some places repeated citation, must be addressed. For the part of Introduction, it looks like citation is everywhere and there are more than 5 papers for each. I think usually it is not necessary to cite at the end of every sentence, you can cite two or three most representative literatures at the end if they are correlated in the content. No need to cite papers as many as you can, especially do not cite things you have not read.
For example, in the section of “1.1. Effects of HIV infection on the Immune System and Intestinal Barrier”, the literatures of citation 1,2,3,4 described the role of the microbiome in people living with HIV. But the author described here is “The gut microbiota is a complex system composed of 10 to 100 trillion bacterial cells, fungi, and viruses.” It does not match with the content, and no need to have 4 citations that expressing similar concept. The following sentence also cited 1,2,3 which is overlapped, the citation 5 and 6 are mis-cited papers. I would suggest combine them as “The gut microbiota is a complex system composed of 10 to 100 trillion bacterial cells, fungi, and viruses. It exists in a symbiotic relationship with the host, particularly the host’s immune system”, and add the appropriate citations.
Line 40: “a major site for HIV replication during the acute phase of infection, and responsible for the majority of CD4 T-cell depletion”. Citations here are not correct. They did not explain CD4 T-cell depletion during HIV infection. As the same suggestion, please combine line 38 to line 40 as a completed writing and add the correct citation at the end. The goal is to make it neat and clear for the readers. Please read the Introduction again line by line to double-check the citations, especially for those cited more than 5 papers, mostly are not accurate and not necessary.
Line 140: regarding the section of “1.5. Effects of HIV infection on the systematic elevation markers”. Citation 3 is shown up at several places. However, citation 3 is a published review, you should avoid secondary citation and cite the original literature. For a trivial example, if you were to state that “soluble CD14 receptors (sCD14), can be measured in the plasma”, you would obviously cite the article that makes that claim, not some review paper that restates it. The same as citation 46 in Line 160 and 161, the paper is not recent, and is not about ART drug and its mechanism. There are so many recently published papers that summarized FDA approved ART drugs that you can cite, such as https://doi.org/10.3390/ph17070887
Minor comments:
1) For Table 1, you can combine Study title and Author as “Villoslada-Blanco et al, Infect Dis Ther. 2022” and “Villoslada-Blanco et al, Sci Rep. 2022”. Is there only two publications discuss the effect of INSTIs on gut microbiota of people living with HIV? Can you please double check and list them all? The same suggestion applies to Table 2.
2) For Table 3, it is hard to read and take the information as you combined INSTI versus NNRTI-based treatment effects on gut microbiota. Can you try to make a graph like below to make it much more understandable? Take a reference of Figure 1C from this paper: https://doi.org/10.1038/s41598-024-68479-4
Author Response
Major comments:
Comment 1: It should be noted that the National Institute of Allergy and Infectious Diseases (NIAID) has updated HIV language guide to eliminate the use of stigmatizing terminology and advance the use of person-first, inclusive, and respectful language. Some of the more commonly used yet most critical terms to avoid are listed as below, I would suggest the authors read the language guide and change the words used in their manuscript: for example, HIV-infected patient (Line 11, 59, 75, 88), HIV-infected individuals (Line 14, 58, 66, 88). I didn’t list them all, please check carefully throughout the manuscript and change it accordingly.
Response 1: Thank you for your language suggestions. In agreement with your feedback, we have carefully reviewed the manuscript and revised the text to ensure more precise and appropriate wording.
Comment 2: The citations in the manuscript need to be improved significantly. The issues such as not appropriate citation, not citing the original work, and in some places repeated citation, must be addressed. For the part of Introduction, it looks like citation is everywhere and there are more than 5 papers for each. I think usually it is not necessary to cite at the end of every sentence, you can cite two or three most representative literatures at the end if they are correlated in the content. No need to cite papers as many as you can, especially do not cite things you have not read.
For example, in the section of “1.1. Effects of HIV infection on the Immune System and Intestinal Barrier”, the literatures of citation 1,2,3,4 described the role of the microbiome in people living with HIV. But the author described here is “The gut microbiota is a complex system composed of 10 to 100 trillion bacterial cells, fungi, and viruses.” It does not match with the content, and no need to have 4 citations that expressing similar concept. The following sentence also cited 1,2,3 which is overlapped, the citation 5 and 6 are mis-cited papers. I would suggest combine them as “The gut microbiota is a complex system composed of 10 to 100 trillion bacterial cells, fungi, and viruses. It exists in a symbiotic relationship with the host, particularly the host’s immune system”, and add the appropriate citations.
Line 40: “a major site for HIV replication during the acute phase of infection, and responsible for the majority of CD4 T-cell depletion”. Citations here are not correct. They did not explain CD4 T-cell depletion during HIV infection. As the same suggestion, please combine line 38 to line 40 as a completed writing and add the correct citation at the end. The goal is to make it neat and clear for the readers. Please read the Introduction again line by line to double-check the citations, especially for those cited more than 5 papers, mostly are not accurate and not necessary.
Line 140: regarding the section of “1.5. Effects of HIV infection on the systematic elevation markers”. Citation 3 is shown up at several places. However, citation 3 is a published review, you should avoid secondary citation and cite the original literature. For a trivial example, if you were to state that “soluble CD14 receptors (sCD14), can be measured in the plasma”, you would obviously cite the article that makes that claim, not some review paper that restates it. The same as citation 46 in Line 160 and 161, the paper is not recent, and is not about ART drug and its mechanism. There are so many recently published papers that summarized FDA approved ART drugs that you can cite, such as https://doi.org/10.3390/ph17070887
Response 2: Thank you for your insightful comment and suggestions. After thorough review, we have revised the citations to ensure greater accuracy in referencing. Additionally, minor adjustments were made to the section discussing the bacterial composition analysis of the HIV infection associated gut microbiome changes, as well as to the potential markers of bacterial translocation and systemic inflammation. These modifications better reflect the newly incorporated references.
Minor comments
Comment 1: For Table 1, you can combine Study title and Author as “Villoslada-Blanco et al, Infect Dis Ther. 2022” and “Villoslada-Blanco et al, Sci Rep. 2022”. Is there only two publications discuss the effect of INSTIs on gut microbiota of people living with HIV? Can you please double check and list them all? The same suggestion applies to Table 2.
Response 1: Thank you for your feedback. We have incorporated a search in Embase, as suggested by the first reviewer, and the updated results are detailed in Figure 1. This search yielded one additional article for PICO2, and we have included its findings in the "Results" section.
Given the distinct aspects addressed by the two articles included in Table 1—one focusing on the bacteriome and the other on the virome—we have determined that it would be appropriate not to combine the two studies. We have revised Table 2 to enhance clarity regarding this distinction.
Comment 2: For Table 3, it is hard to read and take the information as you combined INSTI versus NNRTI-based treatment effects on gut microbiota. Can you try to make a graph like below to make it much more understandable? Take a reference of Figure 1C from this paper: https://doi.org/10.1038/s41598-024-68479-4
Response 2: Thank you for your suggestion. Unfortunately, the proposed graph is beyond our capabilities, as it requires further mathematical analysis of the extracted gut microbiota data from the included studies. Additionally, whole genome sequencing data were not consistently available in the studies to facilitate this analysis. However, we have made Tables 2-4 more transparent to enhance clarity.
Round 2
Reviewer 1 Report
Comments and Suggestions for Authors
The manuscript has been significantly improved.
1. "Furthermore, the effects of prebiotics and probiotics have been extensively researched as potential interventions for disease-associated dysbiosis." - At least some supporting citations are necessary here (citation: pubmed.ncbi.nlm.nih.gov/36986088).
2. "Studies indicate that targeted interventions involving specific microorganisms can alter the complex interactions within the microbiome, leading to health benefits for the host, such as the restoration of dysbiosis to compositions similar to those found in healthy individuals." - Some of the studies have actually found mixed findings.
3. "1. Figure PRISMA flowchart on the study selection process" - Please change this to "Figure 1. PRISMA flowchart on the study selection process".
Comments on the Quality of English Language1. "Beside HIV infection ..." - This should be "Besides HIV infection ..."
Author Response
Comment 1: "Furthermore, the effects of prebiotics and probiotics have been extensively researched as potential interventions for disease-associated dysbiosis." - At least some supporting citations are necessary here (citation: pubmed.ncbi.nlm.nih.gov/36986088).
Response 1: Thank you for your comment. We have incorporated both the suggested citation and an additional one.
Comment 2: "Studies indicate that targeted interventions involving specific microorganisms can alter the complex interactions within the microbiome, leading to health benefits for the host, such as the restoration of dysbiosis to compositions similar to those found in healthy individuals." - Some of the studies have actually found mixed findings.
Response 2: Thank you for your comment. We have updated the text and citations accordingly.
Comment 3: "1. Figure PRISMA flowchart on the study selection process" - Please change this to "Figure 1. PRISMA flowchart on the study selection process".
Response 3: Thank you for the suggestion. We have revised the caption accordingly. Additionally, we identified an error in Figure 1, which has now been corrected in both the text and the figure itself.
Comment 4: "Beside HIV infection ..." - This should be "Besides HIV infection ..."
Response 4: Thank you for your suggestion regarding the language correction. We have updated the text accordingly.